# Selective Hydrogenation Properties of Ni-Based Bimetallic Catalysts

Nobutaka Yamanaka [1] and Shogo Shimazu [2,*]

1   Department of Applied Chemistry, National Defense Academy, 1-10-20 Hashirimizu,
    Yokosuka 239-8686, Japan; yamanaka@nda.ac.jp
2   Department of Applied Chemistry and Biotechnology, Graduate School of Science and Engineering,
    Chiba University, 1-33 Yayoi, Chiba 263-8522, Japan
*   Correspondence: shimazu@faculty.chiba-u.jp; Tel.: +81-43-290-3379

**Abstract:** Metallic Ni shows high activity for a variety of hydrogenation reactions due to its intrinsically high capability for $H_2$ activation, but it suffers from low chemoselectivity for target products when two or more reactive functional groups are present on one molecule. Modification by other metals changes the geometric and electronic structures of the monometallic Ni catalyst, providing an opportunity to design Ni-based bimetallic catalysts with improved activity, chemoselectivity, and durability. In this review, the hydrogenation properties of these catalysts are described starting from the typical methods of preparing Ni-based bimetallic nanoparticles. In most cases, the reasons for the enhanced catalysis are discussed based on the geometric and electronic effects. This review provides new insights into the development of more efficient and well-structured non-noble metal-based bimetallic catalytic systems for chemoselective hydrogenation reactions.

**Keywords:** Ni-based bimetallic catalysts; selective hydrogenation; alkynes; chemoselective hydrogenation; unsaturated carbonyl compounds; unsaturated nitro compounds; reductive coupling

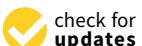

## 1. Introduction

Catalysis has emerged as an important branch of energy and sustainability research because it allows for chemical transformations to be carried out at relatively low temperatures while minimizing or avoiding the formation of byproducts [1,2]. Catalysts can be broadly classified into two groups: homogeneous and heterogeneous catalysts. Homogeneous catalysts have some advantages over heterogeneous catalysts, such as the possibility of carrying out a reaction under relatively mild conditions, higher activity and selectivity, ease of spectroscopic monitoring, and controlled and tunable reaction sites [3]. The main drawback of homogeneous catalysts is the difficulty in separating them from the products after completion of the reaction [4]. Heterogeneous catalysts can overcome this drawback [4]. To date, heterogeneous catalysts based on transition metals have been found to be effective in a number of processes. In particular, hydrogenation is of great importance in petroleum refining and processing and in the manufacture of fine and bulk chemicals [5]. Although most catalytic hydrogenations today rely on precious metals such as Pd and Pt, the high cost and low availability of these metals have caused scientific interest to shift from such precious metals to nonprecious metals for hydrogenation catalysts [6]. Earth-abundant first-row transition metals such as Fe, Co, and Ni have received much more attention due to their specific advantages, such as high abundance on earth, low price, low or no toxicity, and unique catalytic properties [7]. Ni has a long history in the field of catalysis, and its first application for hydrogenation led P. Sabatier to earn the Nobel Prize in chemistry in 1912 [8]. Therefore, Ni is a fascinating alternative to precious metals such as Pd and Pt. However, the chemoselective hydrogenation of a target functional group in the presence of other reactive functional groups in a molecule is difficult to achieve because most transition metal catalysts cannot recognize and preferentially interact with the target group [9].

For this reason, great efforts have been made to seek heterogeneous Ni-based catalysts with high activity for chemoselective hydrogenation reactions.

Effective approaches reported so far include the modification of Ni metal with organic modifiers, reducible metal oxides, and other metals [10,11]. The addition of organic modifiers to the reaction mixture requires subsequent tedious and costly separation. To ensure the occurrence of strong metal-support interactions (SMSIs), catalyst preparation remains critical; for example, the amount of partially reducible metal oxides present must be precisely controlled. The alloying with an additional metal to form intermetallic compounds or alloys will offer an opportunity to design new catalysts with improved activity, selectivity, and stability because the surface atoms have different electronic and geometric structures from those of their monometallic counterparts [12,13]. In the early 1960s, a study on bimetallic catalysts was initiated by J. Sinfelt who worked at Standard Oil Development Company, known today as ExxonMobil Research and Engineering Company, and bimetallic catalysts have continued to attract growing attention in the past decade [14,15]. Alloys are classified into two groups, solid-solution-based and intermetallic-based alloys, depending on their structures (Figure 1) [16]. Solid-solution alloys, in which different metals are randomly mixed at the atomic scale, generally form one of three basic crystal structures: body-centered cubic, face-centered cubic, or hexagonal close-packed structures [17]. The crystal structure of solid-solution alloys changes depending on both their constituent elements and composition [17]. Solid-solution alloys are classified into two groups: substitutional and interstitial solid-solution alloys (Figure 1a,b) [18]. The difference is whether the atoms of the counterpart metal replace or squeeze between those of the parent metal. Intermetallic compounds are composed of different metals with an ordered structure and often a well-defined and fixed stoichiometry (Figure 1c) [19].

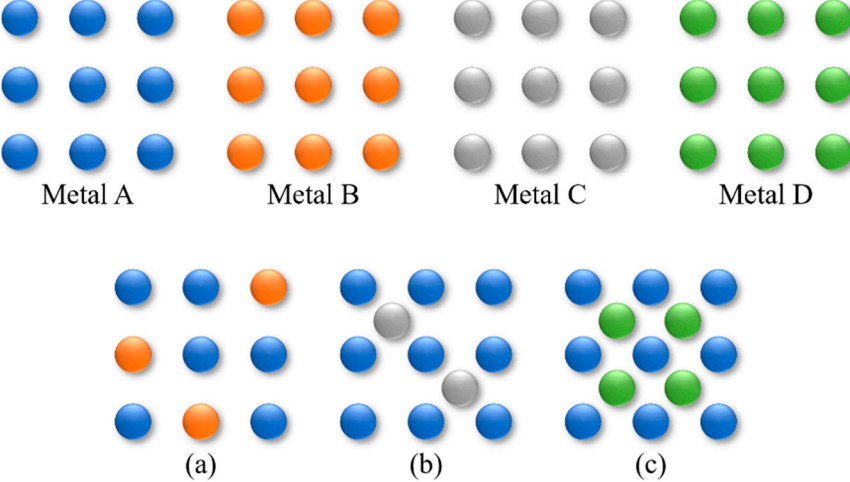

**Figure 1.** Bimetallic alloy structures: (**a**) substitutional and (**b**) interstitial solid-solution alloys and (**c**) intermetallic compounds.

While the currently reported reviews on Ni-based heterogeneous catalysts mainly describe the reforming and decomposition of methane, to the best of our knowledge, there are few available reviews dealing with Ni-based bimetallic alloy-catalyzed hydrogenation reactions [20–24]. Therefore, we hope that this review will provide useful information for future research on the rational design of high-performance non-noble metal-based bimetallic catalysts for chemoselective hydrogenation reactions.

## 2. Preparation Methods of Ni-Based Bimetallic Nanoparticles

There are many well-established methods for preparing bimetallic nanomaterials. The choice of method plays an important role in determining the surface and bulk structure of bimetallic systems [12,25]. This section provides a better understanding of the

relationships between the preparation methods and structural characteristics of Ni-based bimetallic nanoparticles.

Impregnation (co-impregnation and successive impregnation) is a classical and simple method used for the preparation of supported bimetallic nanoparticles and involves drying, calcination, and, finally, reduction [26,27]. The only difference between the two impregnation methods is whether two metal precursors are loaded on a support simultaneously or successively [26]. Sn- or Fe-containing Ni-based alloy nanoparticles were prepared by the coprecipitation method [28,29]. However, in most cases, the reduction of two metal precursors does not occur at the same time because of the difference in their reduction potentials [30]. Successive impregnation is often adopted to prepare core-shell nanoparticles such as $Ni_{shell}Cu_{core}$ and $Cu_{shell}Ni_{core}$ [31]. Generally, the core is composed of a less active metal, and the other active metal is deposited on it.

The following methods enable high phase purity of desired bimetallic alloys. Surface organometallic chemistry on metals (SOMC/M) and chemical vapor deposition (CVD) are similar methods involving the reaction between a supported transition metal and an organometallic [32–35]. These methods enable the controlled deposition of the second metal precursor on the pre-reduced monometallic catalyst but not the support. In the SOMC/M method, the previously reduced monometallic catalyst reacts with an organometallic solution in a paraffinic solvent such as *n*-heptane or *n*-dodecane [32,33]. In contrast, the CVD method introduces the second metal precursor to the surface of the parent metal in the form of a vapor [34,35]. In both cases, hydrogen treatment at an appropriate temperature is finally carried out to form a bimetallic phase with the loss of all the organic fragments. In addition to the CVD method, Komatsu and coworkers reported the synthetic method of Ni-Sn intermetallic compounds from Ni and Sn powders [36]. The mixture of the two metal powders was loaded in an alumina boat in a SiC electric furnace and then melted by raising the temperature from room temperature to 1733 K under flowing argon. This method, which is called arc melting, has disadvantages, such as high energy consumption compared to other methods, although its procedure is much simpler. Polyol-mediated process involves liquid-phase reduction. Various polyols from low-weight ethylene glycol to high-weight polyethylene glycols have several features, such as (i) high boiling points (up to 593 K), (ii) reductant, and (iii) capping agent [37]. In the preparation of Ni-Sn and Ni-Fe bimetallic systems, tetraethylene glycol and polyethylene glycol were used as the solvent and dispersing media, respectively [38,39]. Li et al. prepared Ni-Co and Ni-Cu alloys via a hydrothermal method with hydrazine a reducing agent [40]. The pH and the temperature were the key factors to influence the reactions. Pure alloy powders can be formed only when the pH $\geq$ 13 and the temperature is more than 393 K. Our group developed Ni-based intermetallic compounds via a hydrothermal method without any reducing reagents, followed by a typical hydrogen reduction at high temperatures [41–43]. Table 1 summarizes the second metals and reducing agents used in each preparation method.

**Table 1.** Preparation methods of Ni-based bimetallic nanoparticles.

| Method | Second Metal | Reducing Agent | Reference |
|---|---|---|---|
| Co-impregnation | Sn | $H_2$ | [28] |
| Co-impregnation | Fe | $H_2$ | [29] |
| Successive impregnation | Cu | $H_2$ | [31] |
| SOMC/M | Sn | $H_2$ | [32] |
| SOMC/M | Zn | $H_2$ | [33] |
| CVD | Sn | $H_2$ | [34] |

**Table 1.** *Cont.*

| Method | Second Metal | Reducing Agent | Reference |
|---|---|---|---|
| CVD | Ge | H$_2$ | [35] |
| Arc-melting | Sn | - | [36] |
| Hydrothermal method without reducing reagents | Sn | H$_2$ | [41] |
| Hydrothermal method without reducing reagents | Fe | H$_2$ | [42] |
| Hydrothermal method without reducing reagents | In | H$_2$ | [43] |
| Polyol-mediated process | Sn | NaBH$_4$ | [38] |
| Polyol-mediated process | Fe | H$_2$ | [39] |
| Hydrothermal method | Co | Hydrazine | [40] |
| Hydrothermal method | Cu | Hydrazine | [40] |

## 3. Catalytic Applications of Ni-Based Bimetallic Nanoparticles for Hydrogenation Reactions

This section is divided into four subsections focusing on the hydrogenation of (1) alkynes, (2) unsaturated carbonyl compounds, and (3) unsaturated nitro compounds and (4) one-pot reductive coupling of nitrobenzene and benzaldehyde. Here, we compared the catalytic performances of Ni-based bimetallic catalysts with those of their monometallic counterparts and described the reasons for the enhanced catalysis based on their electronic and geometric effects.

### 3.1. Hydrogenation of Alkynes

Selective hydrogenation of alkynes to alkenes, while avoiding over-hydrogenation to undesired alkanes, represents an industrially important chemical transformation in the manufacturing of polymers as well as fine chemicals [44,45]. For example, the selective hydrogenation of acetylene to ethylene has been used to remove trace acetylene in ethylene feed streams in the production of polyethylene [46]. The most commonly used industrial catalyst for this reaction is based on supported Pd nanoparticles modified by Ag additives, although Lindlar catalyst (Pd poisoned with Pb supported on CaCO$_3$) is not used because of its toxicity [47]. However, this system leaves ample room for improvement, particularly in terms of cost-effectiveness in catalyst design, over-hydrogenation to ethane, and oligomerization to higher hydrocarbons [44,48]. Therefore, it is highly desirable to develop more cost-effective and efficient substituents from both industrial and academic perspectives.

F. Studt et al. used density functional theory (DFT) calculations to determine why Ag showed the high selectivity to ethylene for the hydrogenation of acetylene [48]. The process is as follows: acetylene adsorbs exothermically, and the transition state energies for the first and second steps are below the energy of gas-phase acetylene. The ethylene formed on the surface is subjected to desorption or reaction to undesired ethane. Ethylene from the gas phase can also adsorb on the surface and be hydrogenated to ethane. For PdAg(111), the barrier for desorption is smaller than that of Pd(111). This result explains the reason for the addition of Ag to Pd in the industrial catalyst. These researchers also developed a series of Ni-Zn alloy catalysts on MgAl$_2$O$_4$ spinel supports and evaluated their catalytic performance in the hydrogenation of acetylene in a gas mixture of ethylene, acetylene, and hydrogen [48]. Ni-Zn catalyst with the highest Zn content of 75% showed an even greater selectivity to ethylene than the well-established Pd-Ag system. The DFT calculation for Ni-Zn(110) revealed that there was no obvious difference between the adsorption energy of ethylene and the energy of gas-phase ethylene.

Y. Liu et al. discovered that intermetallic Ni$_x$M$_y$ (M = Ga and Sn) nanocrystals exhibited much higher selectivity for the semi-hydrogenation of acetylene to ethylene than

Pd-based catalysts (i.e., Pd and PdAg) and could be applied to different substrates containing terminal and internal alkynes [44]. A DFT study was carried out to identify why these intermetallic compounds can be used as alternatives to precious metal-based catalysts. Figure 2 illustrates the full potential energy diagram for the semi-hydrogenation of acetylene to ethane on $Ni_3Ga$. The barrier for desorption of ethylene from $Ni_3Ga$ is 0.31 eV, smaller than that from PdAg [48]. More importantly, the transition state energy of ethylene hydrogenation is above the energy of gas-phase ethylene, which indicates that ethylene is subjected to desorption rather than over-hydrogenation to ethane. The excellent selectivity to ethylene in the hydrogenation of acetylene can be assigned to the partial isolation and modified electronic structure of the active metal.

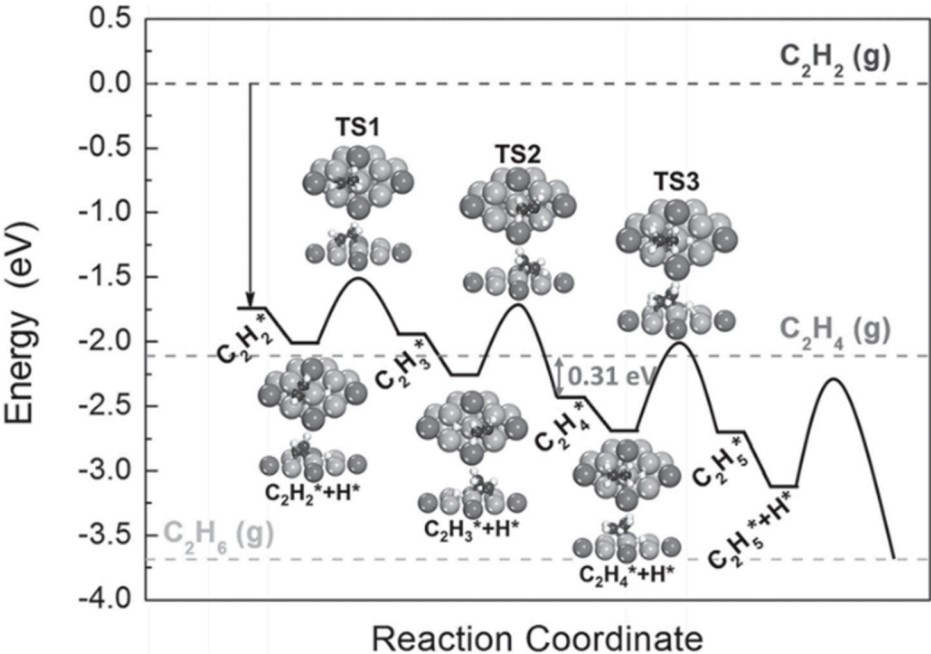

**Figure 2.** Potential energy diagram obtained from DFT calculations for the hydrogenation of acetylene to ethane on (111) of $Ni_3Ga$. The geometries of the reaction intermediates, containing acetylene and ethylene, and transition states along the reaction pathway are displayed in the inset. The Ni, Ga, C, and H atoms are shown by the gray, dark gray, black, and white balls, respectively. Adapted from [44] with permission of Wiley-VCH, copyright 2016.

Komatsu and coworkers studied the catalytic properties of Ni-Sn intermetallic compounds for the hydrogenation of acetylene compared with those of pure Ni [36]. The activity descended in the order of Ni >> $Ni_3Sn$ > $Ni_3Sn_2$ >> $Ni_3Sn_4$. The successive hydrogenation of acetylene into ethane via ethylene seemed to occur over the parent active metal, and ethane was the main final product. However, $Ni_3Sn$ yielded ethylene as the main final product along with only a trace amount of ethane. In the case of $Ni_3Sn_2$, ethane was not observed as a product of the reaction. Therefore, $Ni_3Sn$ and $Ni_3Sn_2$ showed a high selectivity to ethylene for acetylene hydrogenation. These authors presumed, based on the H-D exchange between $C_2D_4$ and $H_2$, that the inhibition of ethylene hydrogenation on $Ni_3Sn_2$ was due to no formation of ethylidyne species by the geometric restriction. The same theoretical prediction might be applied to $Ni_3Sn$. However, it should be considered that the electronic effect might cause the inhibition of ethylene hydrogenation because the electron density of $Ni_3Sn_2$ at the Fermi level was found to be less than that of Ni according to X-ray photoelectron spectroscopy (XPS).

Komatsu's group prepared Ge-containing Ni-based intermetallic compound, $Ni_3Ge$, for the hydrogenation of acetylene and found that it exhibited greater selectivity to ethylene than Ni [35]. This greater selectivity is due to the expanding atomic distance between adjacent Ni atoms and the relatively low electron density of Ni atoms in $Ni_3Ge$. A large

atomic distance will slow the formation of ethylidyne species adsorbed on three-fold Ni sites; ethylidyne is known to be the intermediate in the direct hydrogenation of acetylene into ethane [49]. In addition to the geometric effect, the researchers mentioned the electronic effect on the high selectivity to ethylene. The lower electron density of Ni in $Ni_3Ge$ prevents further hydrogenation to ethane on Ni because there is less back donation [35].

G. X. Pei et al. employed a series of Ag-Ni/$SiO_2$ bimetallic catalysts with varied Ni/Ag atomic ratios of 1, 0.5, 0.25, and 0.125 for the semi-hydrogenation of acetylene in an ethylene-rich stream [50]. When the loading of Ni was relatively high (Ni/Ag = 1 and 0.5), extremely low ethylene selectivity was displayed, similar to that of monometallic Ni/$SiO_2$ catalyst. With the decreased loading of Ni, the ethylene selectivity gradually increased. These authors analyzed the structure of Ag-Ni/$SiO_2$ bimetallic catalysts to determine the reasons for the enhanced ethylene selectivity. Scanning transmission electron microscopy–energy dispersive X-ray spectroscopy (STEM-EDS) analysis of the $AgNi_{0.5}$/$SiO_2$ catalyst with relatively high Ni loading revealed that Ni and Ag were not uniformly distributed in the particles. It is known that it is difficult to form Ag-Ni alloy, and only a small amount of Ni can be alloyed with Ag [51,52]. However, in the $AgNi_{0.25}$/$SiO_2$ catalyst with a relatively low Ni content, most of the Ni interacted closely with Ag because of decreased Ni amounts. Thus, the interaction between Ag and Ni was believed to be responsible for the enhanced ethylene selectivity.

Transition metal silicides have unique chemical properties, such as the lower electronegativity of silicon compared to carbon and the strong modification of the electronic structure around the Fermi level of transition metals [53,54]. However, there have been few reports describing their catalytic applications. C. Liang and coworkers achieved the selective hydrogenation of phenylacetylene to styrene by Ni-Si intermetallics [54]. Adding Si altered the Ni coordination, leading to a strong modification of the electronic structure around the Fermi level compared to metallic Ni; this electronic structure modification influenced styrene adsorption. Ni-Si intermetallic compound prepared by direct silicification at 723 K showed excellent selectivity for styrene (approximately 93%) before the complete conversion of phenylacetylene. Phenylacetylene was very strongly adsorbed on the catalyst surface and blocked the sites for styrene hydrogenation. However, styrene hydrogenation to ethylbenzene occurred when the concentration of phenylacetylene was significantly low. C. Liang group extended the use of Ni-Si intermetallic compounds to other hydrogenation reactions, as shown below.

### 3.2. Hydrogenation of Unsaturated Carbonyl Compounds

The depletion of fossil fuels and environmental deterioration have encouraged a shift from fossil fuel use to biomass resource application [55,56]. Among biomass-derived platform molecules, furfural (FFR) has recently garnered great attention because it can be used as a precursor for many fuel additives and value-added chemicals via many catalytic processes, including hydrogenation, hydrogenolysis, and decarboxylation (Scheme 1) [56–61]. Furfuryl alcohol (FFA) is well-known to be an important hydrogenation product used for the synthesis of fuels and chemicals [55]. Therefore, chemoselective hydrogenation of the carbonyl group in furfural is necessary. However, the hydrogenation of the olefin group is kinetically and thermodynamically favorable compared with that of the carbonyl group [62]. In fact, monometallic Ni catalyst showed low chemoselectivity for FFA because of the formation of a number of byproducts [42].

Ni-Sn intermetallic catalysts (Ni-Sn(X), where X represents the Ni/Sn molar ratios of 3.0, 1.5, or 0.75) were reported to be effective for the chemoselective hydrogenation of FFR to FFA [41]. In particular, Ni-Sn intermetallic phases, $Ni_3Sn$ and $Ni_3Sn_2$, were found to be responsible for the chemoselective hydrogenation of the carbonyl group. The enhanced chemoselectivity compared with monometallic Ni is most likely due to a better interaction of the oxygen lone pair with the partially positive Sn species that are formed by electron transfer between Ni and Sn atoms [63].

**Scheme 1.** Reaction pathway for FFR hydrogenation. Adapted from [61] with permission of Elsevier, copyright 2009.

Ni-In alloy catalysts supported on amorphous alumina (AA) were tested for the hydrogenation of FFR [43]. Ni-In(2.0)/AA contained $Ni_2In$ as the major alloy phase, which quantitatively yielded the desired product, FFA (Conversion >99%, Selectivity 99.9%). Hereinafter, we abbreviate Conversion and Selectivity as Conv. and Sel., respectively. According to X-ray absorption fine structure (XAFS) and DFT calculations by C. M. Li et al., the preferential hydrogenation of the carbonyl group can be explained by the charge transfer from In to Ni [64]. When the hydrogenation of FFA was carried out over Ni-In(2.0)/AA, only 5.2% of FFA was transformed into tetrahydrofurfuryl alcohol (THFA). This result suggests that the addition of In retards olefin hydrogenation by Ni, and consequently, further olefin hydrogenation of the furan ring in FFR did not occur. The decreased adsorption of the olefin group is due to isolation of active Ni sites by In [64].

The catalytic performance of a series of Ni-based bimetallic catalysts (Ni-M(X)HT-Y, where M, X, and HT-Y represent electropositive metals such as Al, Ga, In, Co, Ti, and Fe, the Ni/M molar ratio, and the hydrogen treatment at Y K, respectively) was evaluated for the hydrogenation of FFR [42,65]. As shown in Table 2, modifying the parent active metal with the second metals drastically enhanced the chemoselectivity to FFA. Ni-Fe(2.0)HT-673 catalyst with a higher Ni/Fe molar ratio showed a higher activity for the chemoselective hydrogenation of FFR to FFA (Table 2, entry 8). The catalytic activity of the physically mixed Ni-Fe catalyst was similar to that of Ni HT-673 (Table 2, entries 1 and 9), and the synergistic effect between the two metals was not confirmed. Decreasing the hydrogen treatment temperature to 573 K drastically increased the activity compared to that of Ni-Fe(2.0)HT-673 (Table 2, entries 10 and 11). Fourier transform infrared spectroscopy (FT-IR) study contributed to the elucidation of the mechanism and revealed that FFR has a tendency to adsorb on the Ni-Fe surface through $\eta^1(O)$ configuration [65]. H. Li et al. revealed using XPS analysis that metallic Fe partially donated electrons to Ni [66]. From this result, it can be concluded that the electron-deficient Fe on the Ni-Fe surface attracts the oxygen lone pair of the carbonyl group (Scheme 2). Then, Ni forms hydride-like species through homolytic dissociation of molecular hydrogen, which selectively hydrogenate the carbonyl group.

**Table 2.** Hydrogenation of FFR by various bulk (Ni-M) catalysts. Adapted from [65] with slight modification.

| Entry | Catalyst | Time/min | Conv./% | Sel./% | |
|---|---|---|---|---|---|
| | | | | FFA | THFA |
| 1 | Ni HT-673 | 120 | 100 | 7 | 61 |
| 2 | Ni-Al(1.0)HT-673 | 120 | 100 | 47 | 43 |
| 3 | Ni-Ga(1.0)HT-673 | 120 | 99 | 48 | 24 |
| 4 | Ni-In(1.0)HT-673 | 120 | 20 | 95 | 0 |
| 5 | Ni-Co(1.0)HT-673 | 120 | 85 | 89 | 9 |
| 6 | Ni-Ti(1.0)HT-673 | 120 | 64 | 72 | 20 |
| 7 | Ni-Fe(1.0)HT-673 | 120 | 32 | 97 | 3 |
| 8 | Ni-Fe(2.0)HT-673 | 120 | 74 | 95 | 5 |
| 9 [a] | Ni-Fe(2.0)HT-673 | 120 | 100 | 13 | 75 |
| 10 [b] | Ni-Fe(2.0)HT-673 | 180 | 99 | 96 | 4 |
| 11 [b] | Ni-Fe(2.0)HT-573 | 30 | 90 | 92 | 8 |

Reaction conditions: FFR 1.1 mmol, FFR/Ni molar ratio = 2.0, 2-propanol 3.0 mL, temp. 423 K, $H_2$ 3.0 MPa. [a] Physically mixed Ni-Fe(2.0) catalyst. [b] $H_2$, 1.0 MPa.

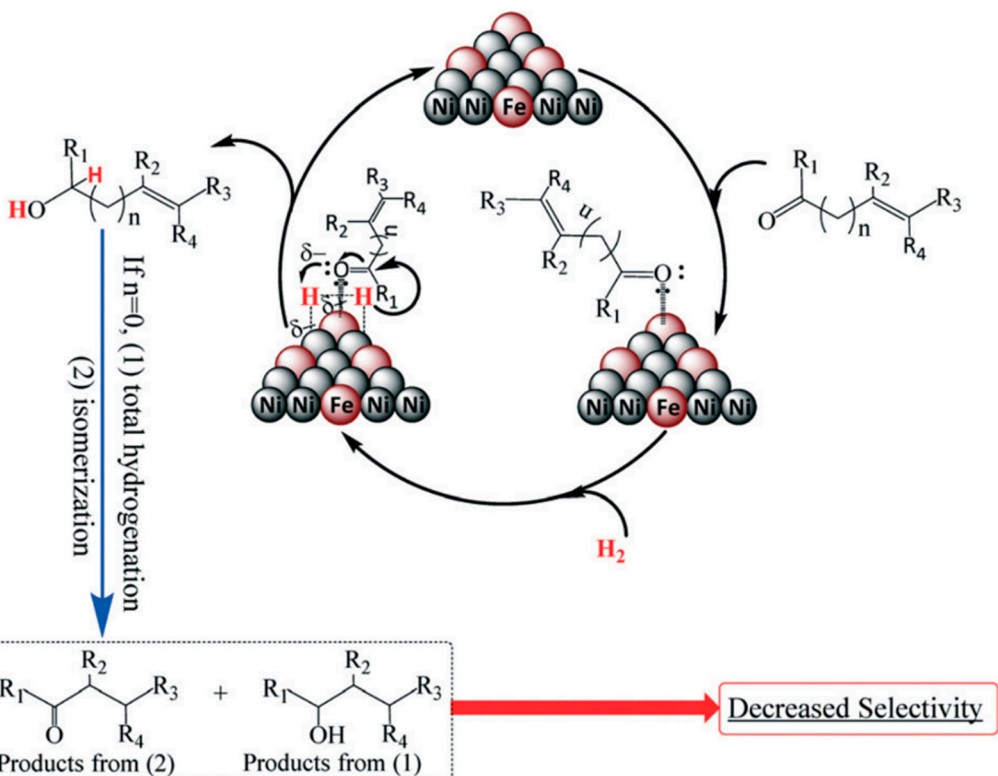

**Scheme 2.** Plausible mechanism for the chemoselective hydrogenation of unsaturated carbonyl compounds to the corresponding unsaturated alcohols by Ni-Fe-based alloy. Adapted from [65] with permission of Royal Society of Chemistry, copyright 2017.

Next, we show a similar change in the adsorption configuration of FFR generated by the addition of a second metal.

In addition to Sn-, In-, or Fe-containing Ni-based alloys, Ni-Cu alloys were also reported to be effective for the chemoselective hydrogenation of FFR to FFA and exhibited better catalytic performances (combined activity/selectivity) than their monometallic counterparts [67]. The alloying of Ni with Cu modified the electronic structure of the two metals, which may have changed the adsorption configuration of FFR on the catalyst surface. The formation of NiCu alloy nanoparticles can increase the adsorption of FFR via $\eta^1(O)$-aldehyde configuration which usually occurs on monometallic Cu sites, while it

can decrease the adsorption of FFR via $\eta^2$(C,O)-aldehyde configuration which is typically observed on monometallic Ni (Figure 3) [68,69].

**Figure 3.** Modes of FFR adsorption on catalyst surfaces. Adapted from [70].

In addition to the chemoselective hydrogenation of unsaturated carbonyls to unsaturated alcohols such as FFR to FFA, the production of saturated carbonyls from their corresponding unsaturated carbonyls also has industrial and biological applications [71]. Cinnamaldehyde (CMA) is an important unsaturated aldehyde that is used in industrial applications, and its selective hydrogenation product, hydrocinnamaldehyde (HCMA), is widely used in the fragrance industry, the synthesis of drugs for AIDS, and the development of other pharmaceuticals (Scheme 3) [71,72].

**Scheme 3.** Reaction network of CMA hydrogenation. Adapted from [54].

The Ni-Si intermetallic compounds prepared by C. Liang group were next investigated for the hydrogenation of cinnamaldehyde [54]. All the Ni-Si catalysts exhibited much higher catalytic performances than metallic Ni catalyst. The improved catalytic activity was due to the abundance of low coordination sites (edge, stepped, and kink) on the catalyst surfaces. The enhanced selectivity for HCMA may be explained by the repulsive force between the silicon atoms in the nickel-silicide intermetallic and oxygen atoms in the carbonyl group of CMA because both silicon and oxygen are electronegative (Scheme 4). Taking advantage of the electrostatic effect, the one-pot tandem synthesis of imines and secondary amines over intermetallic $Ni_2Si/CN$ was designed as described below. The electron shift from nickel to silicon atoms also contributed to the increase in selectivity for HCMA.

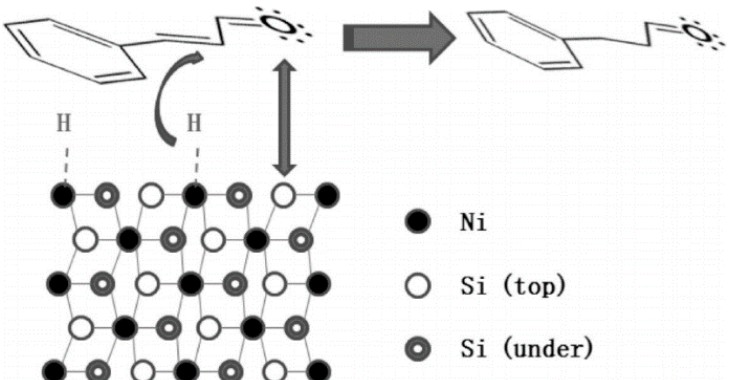

**Scheme 4.** Interaction between CMA and nickel silicide. Adapted from [54].

H. Wang et al. prepared three types of Ni-P/SiO$_2$ alloy catalysts with different Ni/P ratios and reduction temperatures for the selective hydrogenation of CMA to HCMA [71]. The crystal structures were different depending on the preparation conditions. For example, when the Ni/P ratio was 2/1.3, after the hydrogen reduction at 853 K, the X-ray diffraction (XRD) pattern exhibited intense peaks of Ni$_{12}$P$_5$. We denoted this catalyst as Ni$_{12}$P$_5$/SiO$_2$. Increasing the reduction temperature to 1000 K transformed the crystal structure from Ni$_{12}$P$_5$ into Ni$_2$P. We denoted this catalyst as Ni$_2$P/SiO$_2$-1000. Finally, when the Ni/P ratio was 1/1, after the hydrogen reduction at 853 K, the XRD pattern showed not only intense peaks of Ni$_{12}$P$_5$ but also weak peaks of Ni$_2$P. We denoted this catalyst as Ni$_2$P/SiO$_2$-853. The catalytic activity of the Ni$_{12}$P$_5$/SiO$_2$ catalyst was superior to that of Ni$_2$P/SiO$_2$ with Ni/P molar ratios of 2/1.3 (Ni$_2$P/SiO$_2$-1000) and 1/1 (Ni$_2$P/SiO$_2$-853). Ni$_2$P/SiO$_2$-1000 showed a higher activity than Ni$_2$P/SiO$_2$-853. As the only difference between the two Ni-P alloy catalysts was the P content, excess P, which was present in the framework of Ni$_2$P/SiO$_2$-853, covered some of the active sites, which led to decreased activity compared to that of Ni$_2$P/SiO$_2$-1000. In the initial preparation step of Ni$_{12}$P$_5$/SiO$_2$ and Ni$_2$P/SiO$_2$-1000, the same Ni/P molar ratio was used, but their activities were different. C. Stinner et al. found that the shortest Ni-Ni distance of Ni$_{12}$P$_5$ and Ni$_2$P was 2.53 and 2.61 Å, respectively, while that of Ni metal was 2.49 Å [73]. The shorter Ni-Ni distance of Ni$_{12}$P$_5$ indicates that Ni$_{12}$P$_5$ shows more metallic property than Ni$_2$P. This is the most likely reason for the higher catalytic activity of Ni$_{12}$P$_5$/SiO$_2$ than that of Ni$_2$P/SiO$_2$-1000. Therefore, the authors clarified the influences of the P content and crystal structures on the activity of Ni-P alloy catalysts. The selectivity of the three catalysts for HCMA was similar after 480 h, although it was in the order of Ni$_{12}$P$_5$/SiO$_2$ > Ni$_2$P/SiO$_2$-853 > Ni$_2$P/SiO$_2$-1000 at the initial stage of the reaction.

### 3.3. Hydrogenation of Unsaturated Nitro Compounds

Primary amines are useful building blocks in the synthesis of fine chemicals such as polymers, dyes, agrochemicals, and pharmaceuticals [5,9]. Nitro compounds are one of the most abundant starting materials used for the synthesis of primary amines for a long history [74]. The chemoselective hydrogenation of nitro group in the presence of other reducible groups in the same molecule is difficult to achieve; in particular, chemoselective nitro hydrogenation in nitrostyrenes (NSs) is well-known to be the most difficult because the olefin group is more easily hydrogenated than the other reducible groups, including the nitro group [75]. The hydrogenation products are the desired aminostyrene (AS), undesired ethylnitrobenzene (ENB), and over-hydrogenated ethylaniline (EA) (Scheme 5). Here, we focus on the hydrogenation of nitroarenes, including nitrostyrenes, as these are the reactions where Ni-based alloy catalysts can be used.

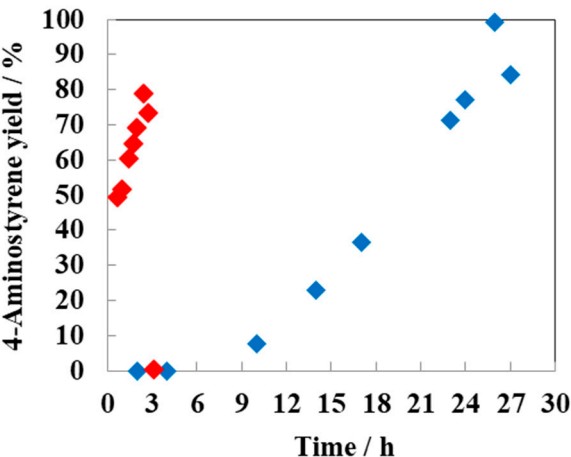

**Scheme 5.** Hydrogenation pathway of 4-NS.

The addition of more electropositive Sn to Ni influenced the hydrogenation pathway of 4-NS [76]. As expected, monometallic Ni catalyst exhibited high activity but low chemoselectivity for the desired 4-AS. The by-production of 4-ENB in the initial reaction stage indicates that the monometallic catalyst could not discriminate between the nitro and olefin groups. The addition of Sn led to a significant improvement in the chemoselectivity for 4-AS. All the Ni-Sn intermetallic compound catalysts preferentially hydrogenated the nitro group over the olefin group, which might be attributed to an interaction between the oxygen lone pairs of the nitro group and the partially electropositive Sn [63]. The $Ni_3Sn_2$ intermetallic compound showed an excellent chemoselectivity of 99% at full conversion after 26 h (Figure 4, blue diamond). When $Ni_3Sn_2$ intermetallic nanoparticles were loaded on a variety of metal oxides, the catalytic activity improved in the order of $Ni_3Sn_2/TiO_2$ > $Ni_3Sn_2/Al_2O_3$ >> $Ni_3Sn_2/ZrO_2$ > $Ni_3Sn_2/SnO_2$ > $Ni_3Sn_2/CeO_2$ [77]. The order was well correlated with $H_2$ uptake but not intrinsic properties (e.g., acidic, amphoteric, basic, reducible, and nonreducible) of the metal oxides used. $TiO_2$-supported $Ni_3Sn_2$ was more catalytically active than unsupported $Ni_3Sn_2$ and produced 4-AS in a remarkably high yield of 79% at full conversion after 2.5 h (Figure 4, red diamond).

**Figure 4.** Time profile for the chemoselective hydrogenation of 4-NS to 4-AS by $TiO_2$-supported (red diamond) and unsupported (blue diamond) $Ni_3Sn_2$.

$SiO_2$-supported Ni-Au alloy nanoparticles with a variety of Ni/Au molar ratios were applied for the hydrogenation of 3-NS and demonstrated higher activity or chemoselectivity for 3-AS than their monometallic Ni/$SiO_2$ counterparts, as shown in Table 3 (e.g., entry 1 vs. entry 4, entry 2 vs. entry 5, entry 3 vs. entry 6) [11]. In general, it is difficult to prepare a homogeneous Ni-Au alloy because of the substantial different reduction potential and immiscibility of the two metals at low temperature [11]. H. Wei et al. used *tert*-butylamine as a weak reducing agent in the second preparation step to form Ni nanoparticles on the Au

surface (Figure 5). Extended X-ray absorption fine structure (EXAFS) results revealed that a large fraction of Ni was alloyed with Au in the $Ni_3Au/SiO_2$ sample after the hydrogen reduction at 550 °C. The enhanced catalysis was because of a synergistic effect between the two metals. Here, it was concluded based on $H_2$ uptake that the formation of the Ni-Au alloy enabled $H_2$ molecules to be activated more easily, improving the catalytic activity for chemoselective nitro hydrogenation. In the Ni-Au bimetallic catalysts, the activity greatly increased with increasing Ni content, while the chemoselectivity remained unchanged at a high conversion of 90% (Table 3, entries 1–3).

**Table 3.** Hydrogenation of 3-NS over various catalysts. Adapted from [11] with permission of Elsevier, copyright 2015.

| Entry | Catalyst | Time/min | Conv./% | Sel./% | $H_2$ Uptake [a] |
|---|---|---|---|---|---|
| 1 | $Ni_{0.33}Au/SiO_2$ | 310 | 90.5 | 96.4 | 12 |
| 2 | $Ni_1Au/SiO_2$ | 160 | 92.5 | 92.1 | 27 |
| 3 | $Ni_3Au/SiO_2$ | 70 | 90.8 | 93.0 | 15 |
| 4 | 0.46%$Ni/SiO_2$ | 960 | 11.2 | 99.2 | 6 |
| 5 | 1.37%$Ni/SiO_2$ | 480 | 17.3 | 97.8 | 3 |
| 6 | 4.11%$Ni/SiO_2$ | 108 | 93.3 | 78.7 | 5 |

Reaction conditions: 0.1 catalyst, 0.5 mmol 3-NS, *o*-xylene, toluene, temp. 50 °C, $H_2$ 300 kPa. [a] µmol/g

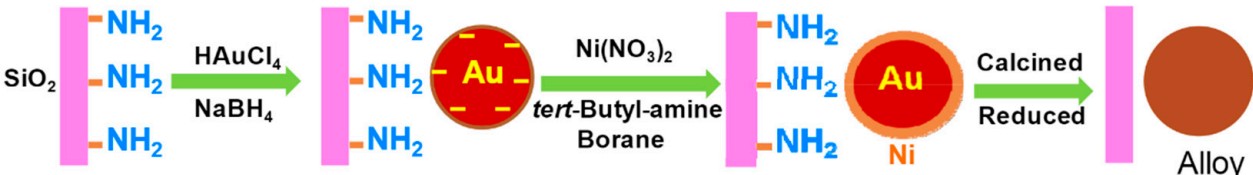

**Figure 5.** Synthetic diagram of Ni-Au alloy formation. Adapted from [11].

A. Corma and coworkers aimed to prepare an active and selective non-noble metal catalyst for the hydrogenation of 3-NS by the introduction of Ni into Co@C NPs [78]. Because of the high ability of Ni to dissociate $H_2$, bimetallic CoNi@C catalyst showed four-fold higher activity than Co@C catalyst while maintaining high chemoselectivity for 3-AS (>97%). The synergistic effect between Co and Ni accounts for the excellent catalytic performance of CoNi@C NPs.

In 2020, a core-shell structured Pt-Ni nanoframe@Ni-MOF-74 (Pt-Ni NF@Ni-MOF-74) showed considerable 4-NS conversion and excellent chemoselectivity for 4-AS (Conv. 100%, Sel. 92%) compared to the individuals, Pt-Ni NF and Ni-MOF-74 [79]. Pt-Ni NF exhibited the highest conversion but low chemoselectivity for 4-AS because it could hydrogenate both functional groups. Ni-MOF-74 exhibited low activity, and it seemed to preferentially hydrogenate the olefin group over the nitro group because of the presence of apical Ni active sites. The slightly decreased conversion in Pt-Ni NF@Ni-MOF-74 relative to that of Pt-Ni NF could be due to the restraint of the Ni-MOF-74 shell on the diffusion rate of 4-NS to the Pt-Ni surface. The high 4-AS chemoselectivity was attributed to the electron transfer from Ni-MOF-74 to Pt suggested by XPS analysis; this electron transfer makes the Pt surface electron-rich and leads to the preferential adsorption of electrophilic nitro group.

In addition to the hydrogenation of NSs, hydrogenation of the simplest nitroarene, nitrobenzene, was employed as a model reaction to evaluate the catalytic properties of Ni-based bimetallic catalysts.

When nitrobenzene was reduced to aniline by Fe metal under anaerobic conditions, a decrease in the reduction rate for nitrobenzene was observed, which was attributed to the precipitation of siderite on the Fe surface, thereby inhibiting the further reduction of an intermediate nitrosobenzene and successive usage [80]. To address this issue, D. R. Petkar et al. prepared Ni-Fe nanoparticles, and their efficiency was evaluated for repeated use in the transfer hydrogenation of nitrobenzene with $NaBH_4$ as the hydrogen source [81]. The catalytic activity was kept constant up to six cycles (Conv. >99%, Sel. 100%). Such

high efficiency was due to the presence of the adjacent Ni sites to suppress the surface corrosion of the Fe site. The Ni sites also facilitated efficient flow of electron transfer from Fe to the adsorbed nitrobenzene and significantly enhanced the catalytic activity of Ni-Fe nanoparticles.

Polymer-supported nano-amorphous Ni-B acted as an effective catalyst for the hydrogenation of nitrobenzene to aniline with hydrazine hydrate as the hydrogen donor [82]. Amorphous alloys have received much attention in the past two decades owing to their physical, mechanical, and chemical properties, which are quite different from those of crystalline alloys of the same composition [83]. Amorphous alloys have small particle sizes, short-range order, and long-range disorder, which provides pathways to novel, more active, and selective catalysts [83,84]. XPS analysis of the sample revealed that the binding energy of elementary B shifted positively compared with that of pure amorphous B, which indicates partial electron transfer from B to Ni. Electron-enriched Ni and electron-deficient B promoted the adsorption of electron-deficient nitrogen and electron-enriched oxygen in the nitro group. The nitro group was then hydrogenated into electron-enriched amine group, which tended to desorb from the catalyst surface.

The two catalytic systems were also suitable for the chemoselective transfer hydrogenation of a variety of substituted nitroarenes to their corresponding aminoarenes. However, nitroarenes containing other reducible groups, such as vinyl and carbonyl groups, were not used in their substrate scope.

*3.4. One-Pot Reductive Coupling of Nitrobenzene and Benzaldehyde*

Imines and their derivatives, such as secondary amines, which are important building blocks used in the manufacture of a variety of functional organic molecules, such as pharmaceuticals, agrochemicals, surfactants, and bioactive molecules, can be synthesized via dehydrogenative condensation between primary amines and carbonyl compounds in the presence of Lewis acid catalysts [85–87]. From the viewpoints of green and sustainable chemistry, one-pot reaction strategies have attracted great interest for their role in increasing the efficiency of chemical synthesis and avoiding intermediate separation and purification steps, thus saving energy and time [88,89]. Here, we report the one-pot chemoselective synthesis of imines and secondary amines from nitro compounds and carbonyl compounds. For the one-pot reductive amination to be successful, not carbonyl compounds but nitro compounds should be reduced first, as shown in Scheme 6 (step I) [90]. Then, the coupling between in situ-formed primary amines and carbonyl compounds occurs rapidly to produce intermediate imines, which are converted to secondary amines through the catalytic hydrogenation of the C=N bond (steps II and III) [90].

**Scheme 6.** Reaction pathway for the reductive coupling of nitrobenzene with benzaldehyde. Adapted from [85].

C. Liang's group carried out the one-pot reductive coupling of nitrobenzene and benzaldehyde to the corresponding arylamines (imine and secondary amine) using an intermetallic $Ni_2Si/SiCN$ catalyst pyrolyzed at 1273 K under argon/hydrogen atmosphere [88]. Such a high pyrolysis temperature allowed metallic Ni to react with the SiCN matrix to form nickel silicide because of the reduction induction SMSI. The SMSI and the formation of intermetallic $Ni_2Si$ modified the electronic structure of the metallic Ni active site, which demonstrated high selectivity for arylamines, mainly the imine. The selectivities of the imine and secondary amine were 82% and 10%, respectively, at >99% conversion of nitrobenzene. Samples prepared at different pyrolysis temperatures (673, 873, and 1073 K) showed no single-phase $Ni_2Si$ in XRD analyses and converted the feedstock of benzaldehyde into benzyl alcohol, resulting in low selectivity for arylamines. To understand the origin of the improved efficiency, the $Ni_2Si/SiCN-1000$ catalyst was subjected to kinetic experiments for nitrobenzene hydrogenation, benzaldehyde hydrogenation, and coupling of benzaldehyde and aniline. The reaction rates of nitrobenzene and the coupling of benzaldehyde and aniline were 1.68 and 3.17, respectively, which were much higher than those of benzaldehyde hydrogenation (0.49). This result is due to the electronic repulsion between the electronegative silicon atoms and carbonyl oxygen atoms [54]. In addition, the active sites of Ni isolated by Si and electron transfer between Ni and Si atoms also contributed to the modification of the metallic Ni active sites, which may have changed the reaction pathway significantly.

D. Esposito and coworkers applied a carbonized filter paper-supported NiFe alloy catalyst for the reductive amination of nitrobenzene and benzaldehyde, which afforded the secondary amine in good yield (Conv. >99%, Sel. 83%), along with benzyl alcohol as a byproduct [91].

In 2021, a more efficient Ni-based bimetallic system was developed for the one-pot synthesis of the imine and secondary amine from nitrobenzene and benzaldehyde [92]. The solvent effect on the catalytic performance of the $Ni_3Sn_2/TiO_2$ intermetallic compound was determined, and a clear relationship between the catalytic activity and solvent polarity was obtained. The catalytic activity was higher in more polar solvent in the following order: protic polar solvent > aprotic polar solvent > nonpolar solvent. Nonpolar mesitylene solvent yielded the imine intermediate in 90% yield without the formation of any byproducts. By prolonging the reaction time, the imine intermediate was successfully converted into the corresponding secondary amine in 80% yield through the catalytic hydrogenation of the C=N bond.

The reductive amination of benzaldehyde with reactive benzylamine was performed using graphene-supported NiPd alloy nanoparticles, which quantitatively yielded the corresponding secondary amine [93]. It is worth mentioning that no side reactions occurred, such as amine dimerization of the reactive benzylamine and hydrogenolysis of the secondary amine [88,94].

D. Esposito's group subsequently used biomass-derivable aldehydes for reductive amination with aminopropanol over the carbon-supported NiFe alloy catalyst [91]. Although amine functionalization of lignocellulose-derived molecules is highly desirable to expand the applicability of biobased chemicals, only a few examples have been reported [91,95]. When furfural and 5-hydroxymethylfurfural were used as the substrates, high conversions and selectivities for their corresponding secondary amines were afforded (Conv. > 99%, Sel. > 78%).

## 4. Conclusions

Through the present contribution, a general overview of the typical methodologies to prepare Ni-based bimetallic systems with different surface and bulk structures and their catalytic applications for the hydrogenation of organic molecules has been offered. Specifically, we featured the selective hydrogenation of unsaturated compounds to desired products (e.g., acetylene to ethylene, furfural to furfuryl alcohol, cinnamaldehyde to hydrocinnamaldehyde, nitrostyrenes to aminostyrenes, and nitrobenzene and benzaldehyde

to the corresponding imine and secondary amine) because monometallic Ni catalyst hydrogenates any reducible functional groups. Ni-based bimetallic alloys and intermetallic compounds showed superior catalytic performances to that of monometallic Ni catalyst. Enhanced catalysis was found to rely on the geometric and/or electronic effects generated by the addition of second metals. We hope that this contribution will provide useful information in this area, and consequently non-noble metal-based bimetallic nanoparticle catalysts showing comparable or higher catalytic performances than those of noble-metal catalysts will be developed in the future.

**Author Contributions:** N.Y. wrote the first draft of this review; S.S. contributed to the writing and revision of the manuscript in its final form. All authors have read and agreed to the published version of the manuscript.

**Funding:** This research received no external funding.

**Acknowledgments:** This work was financially supported by JSPS KAKENHI Grant Number 15K06565 and JSPS Bilateral Joint Research Project (2014–2017) and Frontier Science Program of Graduate School of Science and Engineering, Chiba University.

**Conflicts of Interest:** The authors declare no conflict of interest.

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
