# Peer review of "Selective Hydrogenation Properties of Ni-Based Bimetallic Catalysts"

_2673-4117, doi:10.3390/eng3010006_

Round 1
Reviewer 1 Report
In this review, the authors survey the development of well-established Ni-based catalysts, also provide an instant review on the catalytic applications of Ni-based bimetallic nanoparticles for hydrogenation reactions. In general, this review is acceptable, although some issues still need to be well addressed before publication. Some suggestions are given for reference.
1. In the section of “Preparation methods of Ni-based bimetallic nanoparticles”,the reason for dividing the preparation methods into “impregnation” method and “other” methods is not adequately described. In addition, in “other” methods part,the description of each method is too simple. The last sentence in the first paragraph of the preparation section, "the relationships between the preparation methods and structural characteristics of Ni-based bimetallic nanoparticles", these relations are not actually well revealed in this section.
2. Examples of several alloys for catalytic applications are listed, however, it seems lack of the comparison for different Ni alloys in same application. Further summary or comment on the general rule for choice of Ni-based catalysts to certain reactions are always necessary.
Reviewer 2 Report
The Ni-based bimetallic system was introduced in the review eng-1500628 by Nobutaka Yamanaka and Shogo Shimazu in detail, which involves typical methodologies to prepare Ni-based bimetallic catalyst and catalytic applications of Ni-based bimetallic nanoparticles for hydrogenation reactions. In my opinions, this is an excellent work which may promote the development of more efficient and well-structured non-noble metal-based bimetallic catalytic systems for chemoselective hydrogenation reactions. However, I think there are still some details need to be improved.
1) A lot of pictures and schemes were shown in this review which can help readers understand the content intuitively, but many of them are not clear enough in resolution (Figure 2, Figure 3, Scheme 1, Scheme 2 and so on). For academic review, it is very important to maintain a high and unified resolution. Furthermore, the circles with different colors were used to represent the crystal structures of Bimetallic alloy in Figure 1, but it seems too simple. More complex like 3D images can give readers a more intuitive impression.
2) It can be seen that the author of the review has a deep research on Ni-based bimetallic catalyst fields. The differences of selectivity and activity of catalysts was analyzed from the perspective of geometric and electronic effects in many parts. But there are some ambiguous statements in several parts of the article which interfered with my understanding like the line of 197-203, 210-217 and so on. The author should give the most reasonable explanation through their own professional analysis.
3) Abbreviations are common in academic articles, but I still hope that the author can give detailed names when they appear for the first time so that readers can better understand them. Here are some abbreviations that interfere with my understanding when I read this review, like EXAFS (line 431), 3-NS (line 424).
4) In this review, the fantastic work of many scholars has been reported. Is there any typical example of Ni-based bimetallic catalysts in practical industrial application? Introducing some patents into the review will help broaden the horizon of this review and attract more readers. It would be better if the author could add some applications of Ni-based bimetallic catalysts in industrial production.
Reviewer 3 Report
The submitted paper is a review on hydrogenation reactions using bimetallic heterogeneous nickel complexes. This is a topic area that has not been reviewed yet, and this paper fills that gap.
While this is a review on current developments of bimetallic heterogeneous nickel catalysts, I would suggest adding some historical references. Such as, for the alkyne section (3.1) I think the paper by Brown and Ahuja (J. Org. Chem. 1973, 38, 12, 2226–2230) [https://doi.org/10.1021/jo00952a024] should be referenced.
In figure 2, I would suggest added a third graph with all three potential energy diagrams overlaid on top of each other to better see the energy differences, or just have one graph with all three catalysts systems overlaid.
I found figure 3 very hard to interpret, even when I expanded the screen. I think some color needs to be added to this to make it clearer.
In table 2 I would suggest relabeling the Sel% column, for in its current form it takes some effort to determine what it is conveying. I would suggest Sel% FFA/THFA
